# The Clinical Challenge of Identifying Postural Changes Associated with Musculoskeletal Disorders in a Population of Adolescents: The Evaluation of a Diagnostic Approach

**DOI:** 10.3390/biomedicines12102168

**Published:** 2024-09-24

**Authors:** Roberto Centemeri, Michele Augusto Riva, Michael Belingheri, Maria Emilia Paladino, Marco Italo D’Orso, Jari Intra

**Affiliations:** 1School of Medicine and Surgery, University of Milano-Bicocca, 20900 Monza, Italy; roberto.centemeri@unimib.it (R.C.); michele.riva@unimib.it (M.A.R.); michael.belingheri@unimib.it (M.B.); maria.paladino@unimib.it (M.E.P.); marco.dorso@unimib.it (M.I.D.); 2Department of Occupational Health, Fondazione IRCCS San Gerardo dei Tintori, 20900 Monza, Italy; 3Clinical Chemistry Laboratory, Fondazione IRCCS San Gerardo dei Tintori, 20900 Monza, Italy

**Keywords:** musculoskeletal disorders, somatic dysfunctions, postural dysfunction, MuscleLab, balance test, adolescent

## Abstract

**Background/Objectives**: Inappropriate posture, overweight, and physical inactivity are common causes of pathologies on muscles, ligaments, joints, and bone structures, which could negatively impact the quality of present and future life. The challenge of this work was to develop a diagnostic approach to identify the causes of musculoskeletal disorders in an adolescent population in order to implement preventive procedures. **Methods**: A total of 147 subjects aged between 14 and 18 years who were affected by musculoskeletal disorders and who accessed the Clinical Posturology unit of the IRCCS San Gerardo hospital, Monza, Italy, from 2015 to 2023, were enrolled. The clinical evaluation of each subject included a posturology visit, a physical examination, instrumental devices, such as stabilometric platform, gait analysis, MuscleLab, and imaging tests, such as Radiographic and Magnetic resonance, and a final diagnosis. **Results**: Ninety-eight (66.6%) subjects reported pain at the lumbar spine (33.3%), followed by knee/lower limb (22.4%), cervical spine (13%), and dorsal spine (12.3%). Imaging diagnostics underlined alterations in the musculoskeletal components, bone dimorphism, and asymmetry of the skeleton in 68% of cases. Thirty-one (21%) subjects received a diagnosis of postural dysfunction, seventy-two (49%) received a diagnosis of somatic dysfunction, and ten (7%) received a diagnosis of both postural and somatic dysfunctions. **Conclusions**: Our work highlighted that the three instrumental devices used allowed us to detect somatic and postural functional changes that cause musculoskeletal pathologies in adolescents.

## 1. Introduction

The term “posture” indicates the position assumed by the body in space in relation to the force of gravity. The ideal posture in orthostatic equilibrium is the one that allows for a correct balance of the forces acting on the body, maximum efficiency of movements without pain, and optimal energy economy. The neutral position of the pelvis suggests good alignment of the abdomen, trunk, and lower limbs. Incorrect posture requires greater effort from support structures, leading to an inefficient balance of the body. If maintained over time, this condition can result in symptomatically postural dysfunctions with pain affecting muscles, ligaments, joints, and bone structure. Postural changes can reduce muscular efficiency, thereby generating pathological musculoskeletal conditions. Musculoskeletal disorders are considered the main cause of chronic pain, illness, reduced performance, and quality of life and are responsible for increased irritability, anxiety, depression, disability, and reduced health status. Determining the prevalence of postural dysfunctions is challenging in the general population because of different variables, such as age, weight, work seniority, working hours, number of hours worked per week, time standing or walking, stress levels from work, and exercise habits. Most investigations focused on the working-age population, addressing Work-Related Musculoskeletal Disorders (WMSD) and exposure to biomechanical overload risk [1,2,3,4]. It is known that most postural problems have their origin in childhood and adolescence when the body grows and develops. In fact, there is a consensus that prolonged incongruent posture in childhood and adolescence can cause musculoskeletal disorders in adulthood. Adolescents affected by low back pain have a major risk of suffering low back pain in adulthood. Postural changes affecting the spine in childhood and adolescence increase the risk of degenerative conditions of the spine in adulthood [5,6,7,8,9,10,11]. However, studies on this population reported heterogeneous results because of the diverse research methodologies carried out. In 2007, in a cross-sectional study, Mahlknecht examined postural dysfunction prevalence in subjects aged between 8 and 14 years using the Matthias arm-raising test, revealing a 34% disturbance in the 8–9 age group and a 19% in the 11–14 age group, with no significant differences between genders [12]. In 2008, Cho and coauthors analyzed the prevalence of incongruent posture segmented by body parts in Chinese students, correlating it with psychological distress reported through a questionnaire [13]. Moreover, in 2020, Yang and coauthors estimated the prevalence of incongruent posture in the Chinese school population, analyzing 595,057 children and adolescents. The prevalence was 65.3%, with subjects older than 10 years being more affected than those under 10. Their study also highlighted gender differences, with females being more affected [14]. So far, no systematic data describing the prevalence of postural disorders in the Italian adolescent population have been published. In 2017, the Italian Ministry of Health promoted the development of a document containing clear guidelines for various healthcare professionals involved in the prevention, diagnosis, and treatment of postural disorders, leading to the following series of recommendations: (1) the assessment of postural alignment should involve a standard position; (2) the clinical diagnosis of postural dysfunction requires the evaluation of the alignment between the skull and body segments, as well as the palpation of specific muscle districts and nerve emergence points; (3) the diagnosis of postural dysfunction requires specific clinical evaluations and instrumental investigations to identify its nature; (4) clinical examination of a postural disorder should involve a cranio-caudal approach; and (5) to achieve an improvement in the individual’s health status, treatment should address not only the symptomatic aspects of the soma but also the causal conditions, considering the cranium–caudal correlation [15].

In this work, 147 subjects aged between 14 and 18 years who were affected by musculoskeletal disorders and who accessed the Clinical Posturology unit of the IRCCS San Gerardo hospital, Monza, Italy, from 2015 to 2023, were enrolled to perform and evaluate a diagnostic approach to identify the postural changes that are risk factors in developing musculoskeletal disorders in an adolescent population, with the aim to implement preventive procedures.

## 2. Materials and Methods

### 2.1. Population

A total of 147 adolescents with pain affecting muscles, ligaments, joints, and bone structures were admitted to the Clinical Posturology unit of the IRCCS San Gerardo hospital, Monza, Italy, from 2015 to 2023, for a clinical visit. The inclusion criterion was defined as subjects aged between 14 and 18 years, while the exclusion criterion was stated as individuals that did not complete instrumental assessments.

The local ethics committee waived the requirement for informed consent as all data were de-identified.

### 2.2. Posturology Visit

All the subjects were evaluated by a single posturologist (R.C.) specializing in posturology. The patient’s journey was as follows. (1) First visit: Gathering demographic data, professional activity, and family and medical history. Special attention was given to musculoskeletal conditions, traumas, and previous surgeries. (2) Physical examination: Osteotendinous reflex examination; assessment of scapular plane deviation using a scoliosometer; inspection of lower limbs for apparent discrepancies; podoscopic analysis of foot support; ocular examination, integrating visual acuity data with oculomotor tests; evaluation of jaw occlusion; cranio-caudal assessment of the spine, including shoulder girdle, pelvis, and hip movements; and lower limb examination, including foot, ankle, knee, and hip assessments. (3) Preliminary diagnosis: Based on the examination. (4) Instrumental devices, such as stabilometric platform, gait analysis, surface electromyography, and imaging tests, such as Radiographic and Magnetic resonance, could be conducted to support the diagnosis. 

#### 2.2.1. Somatic Dysfunction Examination

Somatic dysfunction involves the compromise or alteration of related components of the soma, such as skeletal, joint, and myofascial structures, and the corresponding vascular, lymphatic, and neural elements. Diagnosis is based on the following the four criteria summarized in the acronym TART: tissue texture abnormality, asymmetry, restriction of motion, and tenderness, which must be present for the diagnosis [14]. Osteopathic diagnosis and treatment, including somatic dysfunction, are included in the Hospital International Classification of Disease (ICD) [16].

#### 2.2.2. Postural Dysfunction Examination

A postural assessment is a complete examination that involves the study of the whole posture of an individual, following the recommendations of the Italian Ministry of Health document published in 2017 [15]. The assessment included the following: (1) the assessment of postural alignment; (2) the evaluation of the alignment between the skull and body segments; and (3) instrumental investigation tests, which, in our study, included stabilometric platform, surface electromyography, and gait analysis. 

### 2.3. Instrumental Devices

#### 2.3.1. Stabilometric Platform

The stabilometric platform was used because of its fundamental importance in confirming the clinical diagnosis of somatic dysfunction of the spine and evaluating the efficacy of manual therapy, as previously described [17]. The stabilometric platform (SVeP/Standard Vestibology Platform, Politecnica, Modena, Italy) consists of a smooth, hard surface where the patient stands with feet forming an angle of 30°. Briefly, three sensors, forming an equilateral triangle, detect the position of the center of gravity’s ground projection. This information is sent to an electronic processor, representing and monitoring the postural changes over the examination period. The postural oscillations of the patient are qualitatively analyzed, along with the strategies employed to maintain the position. Tests are conducted with open and closed eyes, providing insights into the overall function of the postural system and the role of proprioception. Various test scenarios include the following: (1) occlusal release to evaluate the influence of occlusion and temporomandibular joint (TMJ) function and (2) head rotation to the right and left to evaluate spinal influences.

#### 2.3.2. MuscleLab Test

Initially, a maximal voluntary isometric contraction (MVIC) test was performed on each subject. Measurements were performed on the following muscles: anterior and posterior deltoid, rectus femoris, biceps femoris, medial hamstrings (semimembranosus and semitendinosus), medial and lateral gastrocnemius, and tibialis anterior muscles [18]. After that, the MuscleLab test was performed using surface electromyography (EMG) with linear encoder, which allows us to measure the force and power generated by the active muscles during a movement as a function of time and the amplitude of movement, as previously described [19]. The MuscleLab unit (Politecnica, Modena, Italy) provided inputs for EMG sensors (4000e only) and angle sensors and was connected to a computer with MuscleLab software installed (version 7.18). The electrodes were model T916 (teardrop shape, 43 × 45 mm size, and 4 cm of interelectrode distance) manufactured by Bio Protech Inc. (Wonju City, Gangwondo, Republic of Korea), and hydrogel was used for adhering the electrodes to the skin. Briefly, the electrodes were positioned on rectus femoris (the electrodes were placed at 50% on the line from the anterior spina iliaca superior to the superior part of the patella) and vastus medialis (the electrodes were placed at 80% on the line between the anterior spina iliaca superior and the joint space in front of the anterior border of the medial ligament), following the recommendations for sensor placement of Surface ElectroMyoGraphy for the Non-Invasive Assessment of Muscles project [20]. The test consisted of a repetition of movements, where the subject had to lift one leg and touch a step about 40 cm high with the foot, and then the subject had to bring the foot back to the initial position and carry out the same movement with the opposite leg for 27s at the frequency preferred by the patient. The test was performed in four different modalities to determine the differences between the left and right sides of the human body, as previously reported [19].

#### 2.3.3. Gait Analysis

Gait analysis, a non-invasive examination, is used to assess a subject’s overall walking pattern. In this diagnostic procedure, markers are applied to different joints of the patient, who walks at variable intervals on a treadmill or directly on a platform, as previously described [21]. An optoelectronic system, a video system for gait recording and analysis, and a baropodometric system, measuring foot pressure, are connected (BTS GAITLAB, BTS S.p.A, Garbagnate Milanese, Italy). The examination involves initially analyzing the subject in orthostatic posture and then proceeding to gait analysis. The evaluated parameters included the following: foot contact area with the ground, step length and average speed, potential joint, muscle/tendon, or ligament issues, and neurological disorders. The gait cycle, a cyclic sequence of movements while walking, involves one leg serving as support while the other advances. This cycle includes a stance phase (both feet on the ground) and a swing phase (one foot in the air). The gait cycle is divided into the following intervals: initial double-limb support (10% of the cycle), single-leg load (40%), and terminal double-limb support (10%). These timings vary based on the individual and walking speed.

### 2.4. Variables

The data were entered into a database, including the following parameters: age; reason for the visit; symptoms; previous fractures, major traumas; devices, prostheses (insoles, lifts, corsets) and/or corrective surgery; orthodontic appliance/bite/implant; visual deficiency; sports; stabilometric platform results; surface electromyography results; gait analysis results; X-ray test; magnetic resonance imaging; and final diagnosis.

### 2.5. Statistical Analysis

Statistical analysis was performed using SAS 9.4 (SAS Institute, Cary, NC, USA). The chi-square (χ2) test was used to compare the results obtained in the different groups. A *p* value of less than 0.05 was considered statistically significant.

## 3. Results

This study included 147 subjects, 78 (53%) females and 69 (47%) males. The age ranged from a minimum of 14 to a maximum of 18 years, with a mean age of 15.7 ± 1.2 years and a median age of 16 years. The population was divided into two groups based on the median age, with 68 subjects aged < 16 years and 79 subjects aged ≥ 16 years. The following data on sports activities and current occupation were also collected: 99 (67.3%) subjects practice at least one sport, particularly soccer (*n* = 40), followed by volleyball (*n* = 20) and athletics (*n* = 12).

Ninety-eight (66.6%) subjects presented at least one symptom of pain. The most involved body district was the lumbar spine (33.3%), followed by the knee/lower limb (22.4%), cervical spine (13%), and dorsal spine (12.3%). Less affected by painful symptoms were the ankle/foot (11.6%), shoulder/upper limb (10.9%), and sacral spine (4.8%). Ninety-seven (66.0%) subjects used an orthodontic appliance in the past or at the time of the visit; sixty-two (42.2%) patients presented a visual disorder, such as myopia, astigmatism, or hypermetropia; thirty-eight (25.9%) used insoles, lifts, corsets, or underwent corrective surgeries; and thirty-seven (25.2%) had one or more traumas, fractures, or road accidents.

During the posturology visit, the decision to submit an individual to instrumental tests was based on clinical evaluation carried out by the physician, on a case-by-case basis. Regarding the stabilometric platform, 125 (85%) of the 147 subjects had a positive result, 21 (14.3%) had a negative result, and only in one case, the test was not performed. One hundred and seventeen (79.6%) subjects did not undergo surface electromyography, while out of thirty who underwent the test, twenty-six (17.7%) were positive and four (2.7%) were negative. Regarding gait analysis, 36 (24.5%) patients had a positive result, 5 (3.4%) had a negative result, and 117 (72.1%) did not conduct the test. Imaging diagnostics highlighted an alteration in the structure of the musculoskeletal components, a bone dimorphism, or asymmetry of the skeleton in 68% of cases, whereas 16.3% of subjects were negative, and the last 15.7% did not undergo magnetic resonance tests or X-rays.

After the clinical visit and the results obtained by tests, diagnoses were categorized into different patterns. Thirty-one (21%) subjects received a diagnosis of postural dysfunction with functional alteration of the jaw, foot, or visual axis receptors; seventy-two (49%) received a diagnosis of somatic dysfunction; ten (7%) received a diagnosis of both postural and somatic dysfunctions; seventeen (11.5%) received a negative result, and the last seventeen (11.5%) received a diagnosis different from the ones described above, such as low back and neck pain following sports accidents, dorsal disc disease, bilateral femoroacetabular impingement, low back pain following road trauma, cervical pain, headache, and plantar pain in flat feet (Table 1).

We evaluated the relationship between the type of dysfunction diagnosis (somatic and/or postural) and individual sociodemographic, clinical, historical, and instrumental data. The presence of pathologies involving the dorsal spine, shoulders, and upper limbs was associated with subjects receiving a diagnosis of dysfunction compared with the group of subjects who received a negative or different diagnosis (*p* < 0.05). The presence of pain symptoms was strongly associated with the subjects receiving a diagnosis of dysfunction compared with those who received a negative or different diagnosis (*p* < 0.05). Moreover, a positive stabilometric test was strongly associated with the group diagnosed with dysfunction (*p* < 0.05).

Finally, we observed that the presence of pathologies in the cervical and lumbar spines and in the upper limbs was more frequent in the subjects with a diagnosis of somatic dysfunction (*p* < 0.05). Pain was more present in subjects with a diagnosis of somatic dysfunction (83%), compared with 42% of patients with a diagnosis of postural dysfunction (*p* < 0.05). The present or past use of insoles, lifts, and corrective surgery was more prevalent in the group of subjects with a diagnosis of somatic dysfunction compared with those with a diagnosis of postural dysfunction (*p* < 0.05).

## 4. Discussion

The present study represents the first research conducted in Italy on postural changes that can cause musculoskeletal disorders in an adolescent population. Postural changes are mainly present in the adult population but are also present in children and adolescents. In fact, most postural pathologies originate during body growth and development, particularly in childhood and adolescence, and can negatively impact the quality of life of subjects. Postural changes are related to muscle imbalances due to the long time spent in a specific position, such as inappropriate posture, sitting, lying down, or standing, physical inactivity, and overweight [8,22,23]. The investigation and understanding of the relationships between somatic/postural dysfunctions and musculoskeletal pathologies in adolescents is extremely important since it aids in the identification of people at risk of developing chronic musculoskeletal disorders in adulthood, which limit quality of life and professional activities. Incorrect posture negatively affects the spine and causes muscle imbalances and subsequent pain, thus reducing muscular efficiency and predisposing future musculoskeletal disorders [5,6,7,8,9,10,11]. Despite the limited number, all studies emphasized that the presence of postural changes is the primary cause of musculoskeletal disorders in the adolescent population [8,22,23]. Similarly, our study highlighted a high prevalence of somatic and postural dysfunctions in the adolescents enrolled (77%). However, different parameters and methods were used in the other studies, making comparison difficult. Azevedo and coauthors used the Spinal Mouse^®^ device to evaluate the postural angle in the spinal region and the Namrol^®^ Podoprint^®^ platform to evaluate balance [8]. Boari de Resende and coauthors used the New York scale for the postural assessment, and the classification was made by adding the scores of each test performed [22]. Pacheco and coauthors used Adam’s test, a scoliometer, and the visual pain scale associated with the Nordic Musculoskeletal Questionnaire to determine the prevalence of postural changes and subsequent musculoskeletal disorders in different anatomical regions [6]. Therefore, in our work, we aimed to develop an accurate and standardized diagnostic approach to identify the somatic/postural changes that are risk factors for the development of musculoskeletal disorders in adolescents. First, we identified that there is a strong association between the diagnosis of postural/somatic dysfunction and a positive result obtained using the stabilometric platform test. These data agreed with the national guidelines highlighting that stabilometric platforms are useful in assessing weight distribution, balance parameters, and postural control and, therefore, in the diagnosis of somatic and/or postural dysfunctions [15]. Second, the presence of pain, particularly in the cervical spine, was related to the diagnosis of postural pathology caused by somatic dysfunctions, while the diagnosis of postural pathology without somatic dysfunction was associated with wrong foot support. The relationship between somatic dysfunction and cervical spine pathology might be attributed to the extended periods that adolescents spend in front of a screen, and in a prolonged sitting position, leading to muscular imbalances. Indeed, somatic dysfunction might represent the first risk factor for subsequent postural pathologies up to musculoskeletal disorders; therefore, early identification should be imperative to prevent them. Further research is needed to confirm our data. Finally, performing the three tests in our diagnostic process, i.e., stabilometric, MuscleLab, and gait analysis, allowed us to identify somatic/postural changes that can cause subsequent musculoskeletal pathologies in adolescents. 

Our study presents the following limitations that should be considered: (I) this work is a single-center study, and a larger number of individuals is needed in order to verify and improve the accuracy of our results. (II) The population enrolled belongs to a specific geographical area; therefore, the results obtained may not be extendable to other regions. Future research is needed to determine how socio-psychological, environmental, genetic factors, lack of physical activity, overweight and obesity, improper physical exercise, acute growth periods, imbalanced muscle development, and psychological stress might have played a role. Moreover, a multivariate analysis using all the variables depicted above might be useful to understand the relationships among them, somatic/postural dysfunctions, and the prevalence of musculoskeletal disorders.

## 5. Conclusions

Collectively, in this study, focusing on a population of adolescents allowed us to analyze individuals before future occupational exposures, thus identifying risk factors for the development of musculoskeletal pathologies related to somatic, probably the first step, and postural disorders. If untreated, these pathologies can lead to chronic damage to muscles, ligaments, joints, and bone structures that might manifest in working age, with significant social and economic consequences, as well as a deterioration in the individual’s quality of life.

## Figures and Tables

**Table 1 biomedicines-12-02168-t001:** Data of patients enrolled and laboratory tests for the diagnosis of postural and somatic dysfunctions.

Variable		Postural Dysfunction (*n*)	Somatic Dysfunction (*n*)	*p*-Value
Gender				
	Male	16	31	0.25
	Female	15	41	
Occupation				
	Student	30	71	0.003
	Job	1	1	
Sport				
	Yes	17	53	0.02
	Not	14	19	
Symptoms (pain)				
	Presence	13	60	0.015
	Absence	18	12	
Stabilometric platform				
	Positive	31	64	0.006
	Negative	0	7	
MuscleLab test				
	Positive	7	11	0.03
	Negative	1	1	
Gait analysis				
	Positive	8	18	0.02
	Negative	2	3	

See Section 2 for details. Statistical analysis was performed using SAS 9.4 (SAS Institute, NC, USA), with a significance level set at 0.05.

## Data Availability

The original contributions presented in the study are included in the article, further inquiries can be directed to the corresponding author.

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
