# Peer review of "The Clinical Challenge of Identifying Postural Changes Associated with Musculoskeletal Disorders in a Population of Adolescents: The Evaluation of a Diagnostic Approach"

_biomedicines, 2024, doi:10.3390/biomedicines12102168_

Round 1

Reviewer 1 Report (Previous Reviewer 1)

Comments and Suggestions for Authors

Title: The clinical challenge of the identification of postural changes associated to musculoskeletal disorders in a population of adolescents: the evaluation of a diagnostic approach

In this study, the authors performed a diagnostic approach to identify the causes of musculoskeletal disorders in adolescents. They aimed to reveal that inappropriate posture, overweight, and physical inactivity are common causes of pathologies on muscles, ligaments, joints, and bone structures, negatively impacting the quality of present and future life. The study was conducted on 147 patients who visited the hospital from 2015 to 2023. With all the humility these recommendations are collected with the intention that they can help to improve this work

Line 26: What are the "three instrumental tests"? Does it refer to the physical examination, instrumental devices, and imaging tests? It would be helpful to organize the 2. Materials and Methods section according to the aforementioned sequence if you intend to describe the three instrumental Line 83: Please add information on the reliability and validity of the measurement tools in the Materials and Methods

Line 83: In the Materials and Methods section, please specify the specifications of the experimental equipment (model name, company name, country of manufacture, and year of manufacture).

Line 87: The inclusion and exclusion criteria for the study subjects are too brief. Please describe more appropriate inclusion and exclusion criteria for this study. In particular, using only "aged between 14 and 18 years" as the inclusion criterion seems insufficient. In the end, it appears that the subjects were patients who visited the hospital and were between 14 and 18 years old.

Line 239 (Table 1):What are the criteria for determining postural dysfunction? The research methods do not mention postural dysfunction, and although it is mentioned in the introduction, there is no mention of who diagnosed it, how, and based on what criteria in the actual experiment.

-          Present the mean, standard deviation, and chi-square values.

-          Simply looking at the table, there are more subjects diagnosed with somatic dysfunction examination than those diagnosed with postural dysfunction by experts. What do you think is the reason for this difference, and do you think this difference is clinically significant?

-          What do the Positive and Negative meanings of the Stabilometric platform, MuscleLab test, and Gait analysis signify, and based on what criteria were they determined to be Positive or Negative?

-          According to the Materials and Methods, various variables were examined, but why are the detailed items not analyzed in the result tables?

Preventing postural and musculoskeletal disorders in adolescents is indeed an important clinical challenge. However, aside from the experiment results, various socio-psychological, environmental, and genetic factors such as inappropriate posture, lack of physical activity, overweight and obesity, improper physical exercise, acute growth periods, imbalanced muscle development, and psychological stress might have played a role. It would have been a better study if these factors were also observed during the long study period of nine years.

Author Response

Reviewer 2 Report (New Reviewer)

Comments and Suggestions for Authors

The paper by Centemeri and colleagues entitled “The clinical challenge of the identification of postural changes associated with musculoskeletal disorders in a population of adolescents: the evaluation of a diagnostic approach” aims to develop a diagnostic approach to identify the causes of musculoskeletal disorders in a population of adolescents in order to implement preventive procedures. The authors enrolled a large population of adolescents and performed several instrumental tests, some of which seem to be able to detect somatic and postural functional changes that cause musculoskeletal disorders in adolescents. Overall, the work is interesting ad adds an important contribution to this field of research. However, some issues need to be addressed before publication.

The introduction provides a clear and convincing overview of the importance of posture in adolescents. The authors report that it is widely believed that prolonged incongruent posture in childhood and adolescence can cause musculoskeletal disorders in adulthood (lines 51-52). This part could be considerably deepened and explain which musculoskeletal disorders characterize adulthood with the closest association with postural defects. For example, chronic low back pain is widely prevalent in the adult population. Are postural defects a risk factor?

A total of 147 adolescents with pain in musculoskeletal structures were recruited into the study. Why were pain scores not measured by VAS? Also, the duration of pain should be reported in order to determine whether the pain was acute, sub-acute, or chronic type.

Sports and exercise are known to be effective countermeasures to adolescent musculoskeletal pain and postural defects. The authors could add some additional information regarding this aspect, such as the type of sport/exercise practiced, time spent exercising, and intensity.

The discussion should not be limited to the results obtained but should provide a comparison with the current literature, citing other intervention studies both in agreement and disagreement with the authors' observations.

Author Response

Reviewer 3 Report (New Reviewer)

Comments and Suggestions for Authors

This article seeks to develop a diagnostic method to identify the causes of musculoskeletal lesions associated with physical and postural disorders in adolescents, with the goal of informing disease prevention programs.

However, the manuscript has significant shortcomings, as outlined below: 1. The clinical challenges in identifying postural changes and the background of musculoskeletal disease development in children are not addressed in the abstract. Additionally, the references cited throughout the manuscript are outdated, which undermines the study's novelty and significance.

2. The methodology is questionable. The absence of relevant test charts in the experimental section raises concerns about the study’s accuracy. Furthermore, the research results were not assessed for reliability and validity.

3. The study's sample size is small, with data collected exclusively from adolescent patients in a single hospital, limiting the generalizability of the findings.

In conclusion, I recommend rejecting this manuscript to uphold the high standards of "Biomedicines.

Comments on the Quality of English Language

Appropriate revision of English language and style

Round 2

Reviewer 1 Report (Previous Reviewer 1)

Comments and Suggestions for Authors

Thank you for your thorough Author Response. Your hard work is greatly appreciated

Author Response

Reviewer 2 Report (New Reviewer)

Comments and Suggestions for Authors

The authors fulfilled requests and clarified doubts. 

I have no further comments.

Author Response

Reviewer 3 Report (New Reviewer)

Comments and Suggestions for Authors

the authors has improved the quality of the work.

Author Response

This manuscript is a resubmission of an earlier submission. The following is a list of the peer review reports and author responses from that submission.

Round 1

Reviewer 1 Report

Comments and Suggestions for Authors

Title: A diagnostic approach for the identification of causes of musculoskeletal disorders in a population of adolescents

In this study, the authors performed a diagnostic approach to identify the causes of musculoskeletal disorders in adolescents. They aimed to reveal that inappropriate posture, overweight, and physical inactivity are common causes of pathologies on muscles, ligaments, joints, and bone structures, negatively impacting the quality of present and future life. The study was conducted on 147 patients who visited the hospital from 2015 to 2023. With all the humility these recommendations are collected with the intention that they can help to improve this work

Line 22: "125 (85%) had a positive result at the stabilometric platform, 21 (14.3%) negative" - What does this mean?

-          Stabilometric, Musclelab, and Gait analysis are devices that evaluate presenting symptoms. Isn't it difficult to clearly detect musculoskeletal disorders in adolescents with these?

Introduction: The subject of this study is musculoskeletal disorders in adolescents, but the introduction mainly presents postural dysfunction and its diagnostic criteria. The introduction needs to be logically written to match the subject of this study. For example, information on the prevalence, causes, symptoms, and treatment methods of musculoskeletal disorders among Italian adolescents is needed.

Line 80: In the Materials and Methods section, specify the specifications of the experimental equipment (model name, company name, country of manufacture, and year of manufacture).

-          early present the inclusion and exclusion criteria for study subjects.

-          provide information related to the IRB.

Line 107: Who conducted the somatic dysfunction examination, and how was it conducted? Is it possible to measure myofascial structures and the corresponding vascular, lymphatic, and neural elements with palpation alone?

Line 113: Provide the reliability and validity of the Stabilometric Platform.

-          Specify the specifications and measurement methods of the measurement tools. - experimental equipment (model name, company name, country of manufacture, and year of manufacture).

Line 126: Where were the attachment points of the Rectus Femoris and Vastus Medialis during the EMG test in the MuscleLab test?

-          How was MVIC obtained?

-          Why were the experiments conducted in four different ways?

-          Did you also measure weight changes in conjunction with the Stabilometric during the MuscleLab test conducted with four experimental methods?

Line 144: Provide the specifications, reliability, and validity of the optoelectronic system used in the Gait Analysis.

Line 197 (Table 1):

-          What are the criteria for determining postural dysfunction? The research methods do not mention postural dysfunction, and although it is mentioned in the introduction, there is no mention of who diagnosed it, how, and based on what criteria in the actual experiment.

-          Present the mean, standard deviation, and chi-square values.

-          Simply looking at the table, there are more subjects diagnosed with somatic dysfunction examination than those diagnosed with postural dysfunction by experts. What do you think is the reason for this difference, and do you think this difference is clinically significant?

-          What do the Positive and Negative meanings of the Stabilometric platform, MuscleLab test, and Gait analysis signify, and based on what criteria were they determined to be Positive or Negative?

-          According to the Materials and Methods, various variables were examined, but why are the detailed items not analyzed in the result tables?

Preventing postural and musculoskeletal disorders in adolescents is indeed an important clinical challenge. However, aside from the experiment results, various socio-psychological, environmental, and genetic factors such as inappropriate posture, lack of physical activity, overweight and obesity, improper physical exercise, acute growth periods, imbalanced muscle development, and psychological stress might have played a role. It would have been a better study if these factors were also observed during the long study period of nine years.

Reviewer 2 Report

Comments and Suggestions for Authors

This is not an “Article” that report scientifically sound experiments and does not provide a substantial amount of new information according to “Instructions for Authors”. This may be an “Opinion”. The aim is not suitable for a scientific article, “to perform a diagnostic approach to identify the causes of musculoskeletal disorders” (L13, and L77).  It is just an experience report. The causes of musculoskeletal disorders could not be identified because the sample consisted only of adolescents with musculoskeletal disorders, and the authors did not describe the causes. The participants were unknown for readers, “adolescents who were admitted to the Clinical Posturology unit of the IRCCS San Gerardo hospital”, at least they do not include the control. The issue was not addressed, “it is challenging to determine the prevalence of postural dysfunctions in the general population (L42)”, and “no systematic data describing the prevalence of postural disorders in the Italian adolescent population (L61)”. The measurements, such as stabilometric platform, MuscleLab test, and Gait analysis, did not contribute the estimation of the prevalence of postural disorders or musculoskeletal disorders. The two disorders are confusing. And the measurements did not contribute the identification of the causes (L200216). Only the associations were described. Furthermore, the measurements had no diagnostic criteria (L113157). The concept is disorganized, and the manuscript is incoherent. Moreover, there is an ethical issue. Even when the data were de-identified, informed consent should be obtained because the first author (R.C.) assessed all participants in his affiliation.